# Comparing the learning dynamics of in-context learning and fine-tuning in language models

**Basile Confavreux**
Gatsby Computational Neuroscience Unit
University College London

**Aaditya K. Singh**
Gatsby Computational Neuroscience Unit
University College London

**Jin Hwa Lee**
Sainsbury Wellcome Centre
University College London

**Amaury Sabran**
WaveForms AI

**Andrew M Saxe**
Gatsby Computational Neuroscience Unit
Sainsbury Wellcome Centre
University College London

## Abstract

Pretrained language models can acquire novel tasks either through in-context learning (ICL)—adapting behavior via activations without weight updates—or through supervised fine-tuning (SFT), where parameters are explicitly updated. Prior work has reported differences in their generalization performance and inductive biases, but the origins of these differences remain poorly understood. In this work, we treat ICL and SFT as distinct learning algorithms and directly compare the learning dynamics they induce across medium-sized models, analyzing both the evolution of their inductive biases and the underlying internal representations. We find that ICL preserves rich input representations but imposes stronger priors inherited from pretraining, whereas SFT suppresses task-irrelevant features—potentially explaining its weaker generalization in few-shot regimes. These results highlight a mechanistic distinction between context-driven and weight-driven learning.

## 1 Introduction

Large language models (LLMs) can acquire new tasks either in context (ICL), for instance, by providing example–label pairs at inference time with no weight updates (Brown et al., 2020; Liu et al., 2023), or via supervised fine-tuning (SFT), by changing model parameters typically with gradient-based updates on labeled data (Vieira et al., 2024). While both learning strategies can achieve good performance (Agarwal et al., 2024), mounting evidence indicates they differ in inductive biases, order sensitivity, and out-of-distribution (OOD) behavior, with ICL sometimes generalizing more robustly than SFT even when trained on the same data (Chan et al., 2022b; Lampinen et al., 2025; Akyürek et al., 2022). Understanding how these divergences arise has been difficult in naturalistic settings where task semantics, priors, and data geometry are hard to control.

Here, we treat ICL and SFT as two distinct *learning algorithms*, and compare the learning dynamics they elicit in medium-sized pretrained transformers (Vaswani et al., 2017) ($\geq 8B$ parameters) on a minimal, 2-D linear classification task with semantically unrelated labels (Wei et al., 2023; Agarwal et al., 2024; Min et al., 2022). This setting minimizes confounds from linguistic priors or label semantics present in open-domain tasks, and enables precise control over the task geometry (decision boundary angle), shot count and example ordering, which we use to unveil the generalization strategies at play. We compare ICL and SFT on the same task instance and ordering of examples, and track accuracy, smoothness, confidence, inferred boundary angle, and layer-wise representa-

tional similarity analysis (RSA). Despite both ICL and SFT reaching similar held-out accuracy, we find that ICL exhibits stronger pretraining-inherited priors compared to SFT, biasing the generalization patterns towards specific computations such as number comparison and pattern matching of in-context labels' ordering. Moreover, ICL preserves a rich representation of inputs, whereas SFT suppresses task-irrelevant features and exhibits representation compression/collapse aligned with task labels. These differences manifest in task-angle-dependent generalization, ordering effects, and distinct representational geometries across layers.

Our main contributions are:

- Controlled, head-to-head comparison of ICL vs SFT across matched trajectories, which reveals different inductive biases that manifest in task instance sensitivity, order effects, and confidence profiles.

- Representational analysis of models' internal representation showing that SFT representations collapse by label, while ICL largely maintains input structure across layers.

- Bridging theory and practice: we connect empirical patterns to views of ICL as implicit optimization/Bayesian inference (Von Oswald et al., 2023; Garg et al., 2022; Dai et al., 2022; Zhang et al., 2024) and to recent reports of ICL's superior generalization compared to SFT (Akyürek et al., 2022; Bai et al., 2023; Lampinen et al., 2025; Chan et al., 2022b).

Together, these results provide a mechanistic view into the differences between context-driven and weight-driven learning.

## 2 RELATED WORK

**Many-shot ICL:** LMs can learn high-dimensional numeric functions and semantic tasks directly from long in-context sequences, with performance continuing to improve well beyond few shots (Agarwal et al., 2024; Anil et al., 2024). Here we unpack the scalar performance metrics in one such task to obtain more fine-grained generalization patterns, unveil inductive biases of ICL and compare them to SFT.

**ICL vs SFT generalization:** Many studies have compared the efficiency and generalizability of SFT and ICL. Previous work showed that ICL can out-generalize SFT on a range of tasks, and identified regimes where SFT recovers similar performance through augmentation and regularization (Lampinen et al., 2025). ICL exhibits superior generalization performance on tasks containing implicit patterns, even when providing more data for SFT (Yin et al., 2024), while other studies report better generalization for SFT over ICL in other tasks (Mosbach et al., 2023), suggesting a more nuanced picture. In this work, we compare ICL and SFT across a learning trajectory and correlate the observed differences with the internal representation elicited by ICL and SFT.

**Representations under SFT and/or ICL:** SFT is known to compress representations towards task-relevant directions (Kumar et al., 2022). Previous work compared the representations for ICL vs SFT in a semantic-heavy task (MMLU), and reported that SFT elicited more task-aligned representations than ICL (Doimo et al., 2024). However, they did not unpack learning dynamics, i.e., the influence of the progression of in-context examples.

**Ordering and selection effects in ICL:** Demonstration order strongly affects ICL, with early/last examples disproportionately influential; mitigations include representative/active selection and calibration (Zhang et al., 2022; Yang et al., 2023). We extend these findings with periodic-pattern probes that induce rule-following over feature-use in some cases.

**Mechanisms: ICL as implicit optimization/Bayesian inference:** Theoretical accounts link ICL to Bayesian inference or implicit gradient descent under pretraining distributions and architecture constraints (Bai et al., 2023; Garg et al., 2022; Dai et al., 2022; Akyürek et al., 2022). Follow-ups caution that such mechanisms do not necessarily translate to larger models trained on naturalistic data (Shen et al., 2023; Raventós et al., 2023), aligning with our observations on medium-sized LMs.

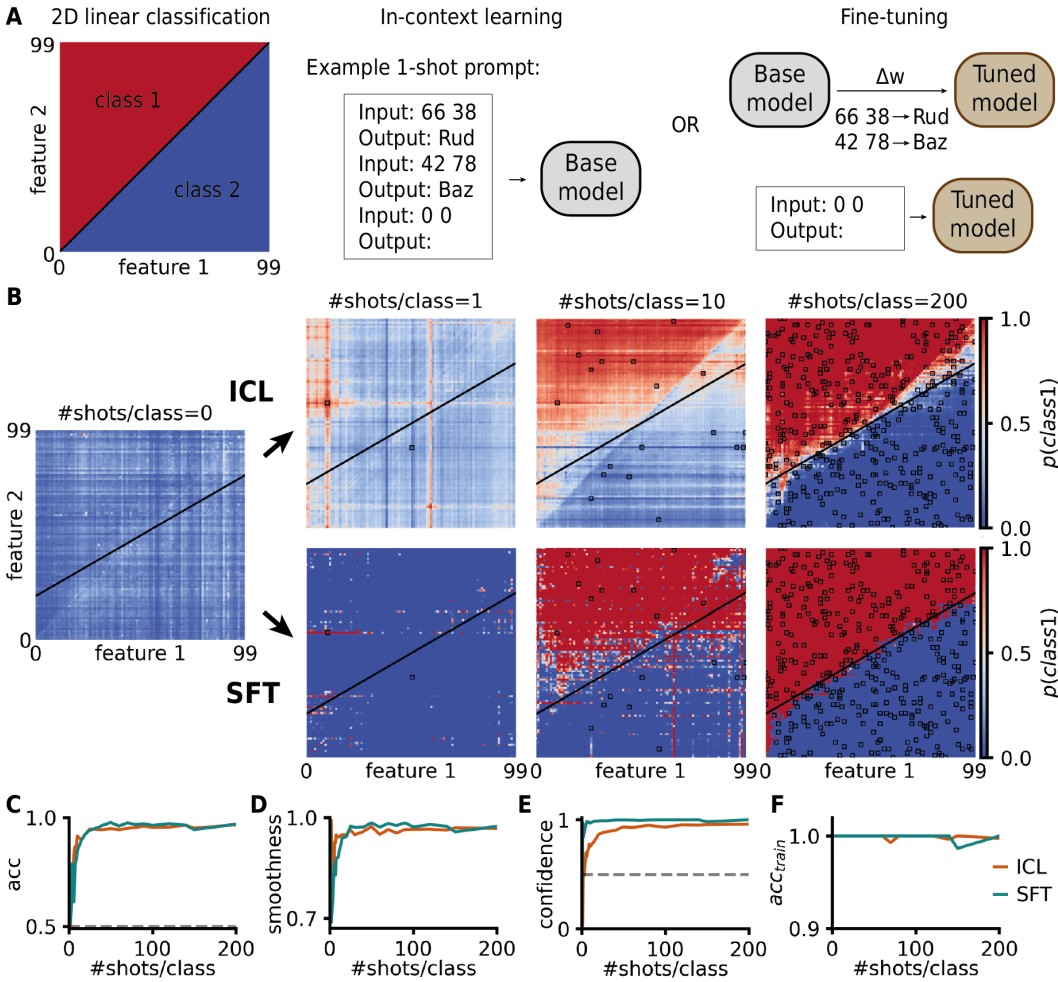

Figure 1: **Decision boundaries for ICL and SFT on a 2-D linear classification task. A**: Description of the 2-D linear classification task: the inputs are two integers $(n_1, n_2) \in \mathbb{N}^2, n_1, n_2 < 100$ and the outputs are two classes with semantically unrelated labels "␣Baz" and "␣Rud". The model was trained on this task either using ICL or SFT. **B**: Two example trajectories (one for ICL, one for SFT) on the same instantiation of the task (same training set and ordering of examples at each shot. For 0, 1, 10 and 200 shots per class, the probability associated with the logit of class 1 for all 100 x 100 possible inputs in the task. The probabilities are normalized for decision making such that $p(class\ 1) + p(class\ 2) = 1$. The black line denotes the ground-truth decision boundary ($\theta = 30°$). Black squares indicated the examples present in-context (ICL) or in the training set (SFT). **C-F**: Evolution of the accuracy (C), smoothness (D), confidence (E) and training accuracy (F) for the two learning trajectories shown in B, as a function of the number of shots per class. The smoothness is defined as (1 - the fraction of model outputs that have 2 or more neighbors of the opposite class).

## 3 METHODS

### 3.1 TASK: 2-D LINEAR CLASSIFICATION

We considered a 2-D linear classification task with single-token inputs and outputs, adapted from previous work (Agarwal et al., 2024) (Fig. 1A), which showed meaningful performance improvements in the "many-shots" regime (i.e. hundreds to thousands). Concretely, each instance of the task defined a linear decision boundary over ordered integer pairs $\mathbf{x} = (n_1, n_2)$, with $n_1, n_2 \in \{0, \ldots, 99\}$. Models had to map inputs to one of two labels (e.g., "␣Baz"/"␣Rud"). The task was parameterized by a single parameter $\theta \in [0, 180°]$, the angle of the ground-truth decision boundary

relative to the first feature $n_1$ (Fig. 1A). Note that all versions of this task had a balanced dataset, i.e., the same number of examples for each class, at every shot count.

## 3.2 SEMANTICALLY UNRELATED LABELS

To minimize verbalizer priors, we avoided common placeholders ("Foo"/"Bar") and selected label tokens that were single-token under most open-source tokenizers and less frequent in pretraining corpora ("␣Baz"/"␣Rud", see Supp. Fig. A13 for other label choices).

## 3.3 MODELS

Our primary model was Llama3-8B (Dubey et al., 2024) (Fig. 1, 2 & 3). We replicated key experiments across different model families and sizes: Qwen3-8B (Yang et al., 2025), Gemma3-12B and 27B (Kamath et al., 2025), Qwen3-0.6B,1.7B,4B, as well as gpt-oss-20B (Agarwal et al., 2025) (Fig. 4 & Supp. Fig. A3).

## 3.4 PROTOCOLS: ICL AND SFT

**ICL:** Prompts contained $K$ randomly sampled exemplars per class ("$K$ shots/class") drawn without replacement from a pre-generated training pool, followed by a single query (Fig. 1A). We study the same ordered stream of examples across shot counts to form learning trajectories. When analyzing ordering effects, we either generate new pre-generated training pools (Fig. 2A-H), impose periodic patterns (Fig. 2I-L) or shuffle in-context ordering as controls (Supp. Fig. A1).

**SFT:** We trained on the same cumulative dataset and ordering as ICL at each shot count. Unlike ICL, which does not have explicit hyperparameters, we had to choose several hyperparameters for the fine-tuning. Unless specified, we used the AdamW optimizer with a cosine learning rate schedule. We report hyperparameters and stability analyses in Appendix (see Supp. Fig. A2 for additional details).

**On the term "learning dynamics":** For ICL, different "shots" correspond to independent prompts with progressively more examples; there are no weight updates. We use *learning dynamics* as a convenient shorthand for performance and representation changes as the in-context dataset grows. In SFT, shots per class index the same cumulative dataset used for training, though the base model is trained from scratch for every shot count on the relevant training examples. We believe this is a useful abuse of notation as it enables us to compare ICL and SFT as two *learning algorithms*.

## 3.5 METRICS

For each learning trajectory, we tracked the (i) *accuracy* on all $100 \times 100$ possible inputs to the task, (ii) *smoothness*, defined as 1 minus the fraction of grid points whose predicted class disagrees with at least two of their four neighbors, (iii) *confidence*, measured as the maximum softmax probability, and (iv) *inferred angle*, obtained by fitting a linear classifier to the model's predicted labels on the grid.

## 3.6 REPRESENTATIONAL SIMILARITY ANALYSIS (RSA)

We computed cosine-similarity matrices of last-query-token activations across (i) all prompts along each trajectory and (ii) a mixed set of training and randomly sampled test inputs (Kriegeskorte et al., 2008). The activations were collected after the MLP at every layer in Llama3-8B (32 layers). We summarized layer-wise patterns and compared ICL vs SFT at matched shot counts.

The companion code is accessible at `https://github.com/basile6/ICLvsSFT`

# 4 RESULTS

We compared how medium-sized pretrained language models (LMs) learned a novel task either in-context (ICL) or via supervised fine-tuning (SFT), matching the two procedures on the same

training items, order, and shot counts in a 2-D linear classification task. We analyzed generalization performance across shots and task instances, in tandem with layer-wise representational similarity analysis (RSA) to unveil differences in representations and inductive biases.

## 4.1 SIMILAR GENERALIZATION PERFORMANCE WITH DIFFERENT INDUCTIVE BIASES

Having defined our 2-D linear classification task (Methods & Fig. 1A), we first verified that Llama3-8B could solve this task both with ICL and with SFT (Fig. 1B). Under matched data and training examples ordering, held-out accuracy was similar across learning trajectories, with similar speeds of learning (Fig. 1C). Both approaches also achieved near-perfect training accuracy throughout the learning trajectory (Fig. 1F). However, SFT consistently yielded higher confidence than ICL at comparable shots (Fig. 1E), suggesting stronger alignment of logits with the task labels. We also verified that the ICL behaviour was robust across seeds and in-context shuffling of examples (Fig. A1).

The decision fields revealed qualitative differences in inductive biases (Fig. 1B). Especially for few shots, ICL showed (i) a "previously-seen feature value bias", extrapolating along rows/columns that reuse values shown in-context, and (ii) a "comparison bias" that favors decision boundaries near the diagonal ($\theta \approx 45°$), consistent with "which number is larger?" heuristics (Fig. 1B, 1 and 10 shots/class). These biases remained detectable even when global accuracy had converged (Fig. 1B, 200 shots/class).

## 4.2 QUANTIFYING INDUCTIVE BIASES BY VARYING TASK ANGLE

To expose the inductive biases observed in Fig. 1 more quantitatively, we compared model performance across learning trajectories for several task angles $\theta$. In principle, all these task instances were of similar difficulty. However, we hypothesized that the "previously seen feature value bias", which induced row and column generalization (considering the task representation introduced in Fig. 1), would translate into better performance for $\theta = 0°$ and $\theta = 90°$, which are aligned with these generalization patterns, compared to other task angles. Conversely, the "comparison bias" suggested $\theta = 45°$ as another favored angle. Both predictions were verified when comparing model performance across seeds for ICL (Fig.2A,B,D). Moreover, when inferring the optimal linear classifier from the model output (Fig.2C), we observed an overestimation (resp. underestimation) of the inferred task angle for $\theta = 30°$ (resp. $\theta = 60°$), consistent with a diagonal pull from the comparison bias. This could already be seen from the fine-grained generalization behaviour shown in Fig. 1B. SFT was not bias-free either under this probing with various task angles (Fig. 2E-H), and displayed increased performance for the "easier" angles (similarly to ICL, $\theta = 0°$, $\theta = 45°$, $\theta = 90°$), but not as strong a diagonal pull as ICL (Fig. 2G,H).

## 4.3 ORDERING EFFECTS AND PATTERN-INDUCED RULE FOLLOWING

We noticed that the ordering of the training examples had an effect in ICL if there was a pattern, i.e. a period, in the ordering of the labels. For instance, always showing a class 1 example before a class 2 example prompted the model to output that all following queries were of class 1, irrespective of their features—including those provided in context to be of the other class. In this case, the model ignored the input feature values and instead followed the logic of pattern matching, and not the one of linear classification (Fig. 2I,J). This behaviour was consistent across all 10 randomly-sampled, balanced learning trajectories.

However, longer-period patterns (e.g., "12121221") exerted smaller or no detectable influence on held-out accuracy compared to the random case, suggesting a short-horizon sensitivity to label interleaving (Fig. 2I,K,L). It thus appeared that LMs could implement not only fixed ICL rules, but select among algorithms in-context, such as pattern-matching, previously-seen feature generalization, number comparison and linear classification. This finding confirms what had been proposed previously in a more theoretical setting (Bai et al., 2023), and such strong rule-based generalization patterns match previous empirical reports in medium-size transformers (Chan et al., 2022b).

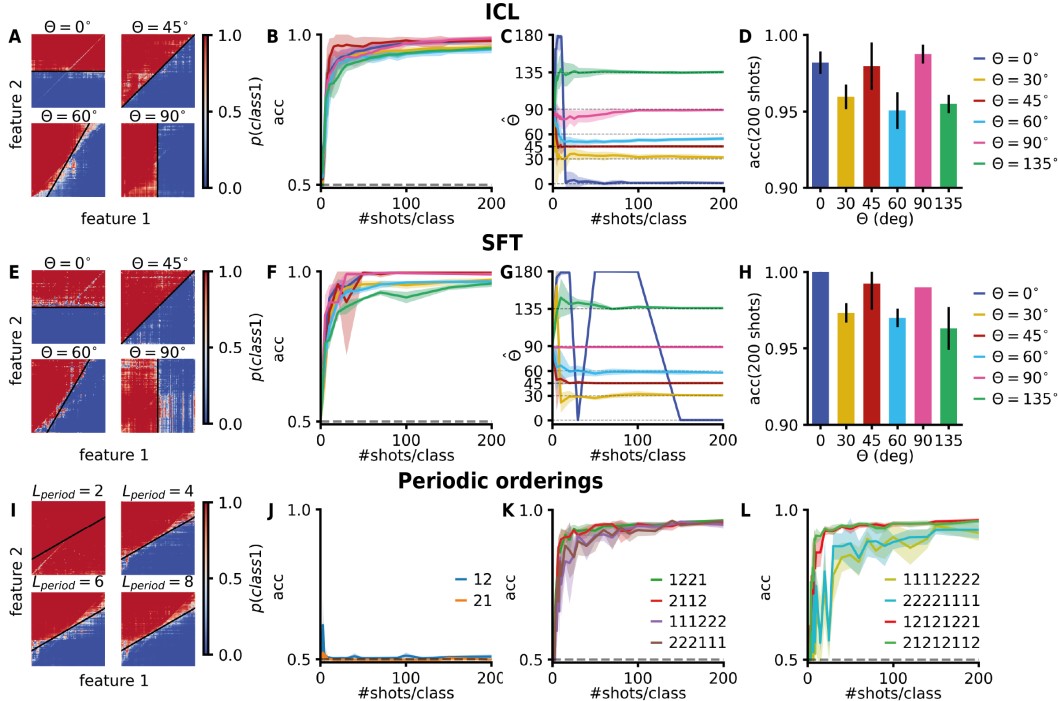

Figure 2: **Quantifying inductive biases. A**: Model decision boundary after 200 shots/class with ICL (Llama3-8B) on four task angles ($0°$, $45°$, $60°$ and $90°$). **B**: Evolution of the accuracy across shots/class for different task angles (mean and standard deviation, computed across 20 seeds per task angle). **C**: Evolution of the angle of the optimal linear classifier angle inferred from model outputs (mean and standard deviation, computed across 20 seeds per task angle) **D**: Accuracy for 200 shots/class (mean and standard deviation, computed across 20 seeds per task angle) for different task angles. **E-H**: Same as A-D but for SFT (Llama3-8B). **I**: During ICL learning trajectories, ordering the examples in-context with a pattern. The training sets are still balanced. "12" corresponds to the strict alternation of class 1 and class 2 examples provided in-context. More complicated sequences with longer periods are also considered (e.g. "12121221" of length 8). Visualization of the model output for all 10,000 task inputs, as in A and B, but for periodic orderings of different period lengths ("12", "1221", "121221" and "12121221") **J-L**: Evolution of the accuracy across shots/class for different periodic orderings (mean and standard deviation computed over 10 trajectories).

## 4.4 INTERNAL REPRESENTATIONS: SFT COLLAPSES REPRESENTATIONS ALONG LABEL AXES, ICL PRESERVES STRUCTURE

We wondered whether the differences in inductive biases observed above for ICL and SFT translated to differences in the internal representations of the model. For each of the 10,000 task inputs, we extracted the activations at each layer of the model for several shot values along the same trajectory for SFT or ICL. We then computed the cosine similarity between layer-wise activations of all inputs by the model for all layers to obtain representational similarity matrices (Kriegeskorte et al., 2008). At 200 shots/class, with both ICL and SFT achieving similar training and generalization performance, substantial differences could be seen in the model representations between ICL and SFT. Although the representations in early layers were similar (Fig. 3A,B), by the middle layers, SFT had elicited what appeared to be a collapse of the representations alongside the task labels (Fig. 3B). In other words, the activations clustered in two opposite directions, one for class 1 and the other for class 2 examples. In contrast, ICL maintained more varied input-specific representations throughout all layers (Fig. 3A).

We also investigated the evolution of the representations in one layer for increasing numbers of shots (Fig. 3E). A major difference beyond the representation collapse already observed above was the representation of task examples from the training set. For ICL, examples present in the training

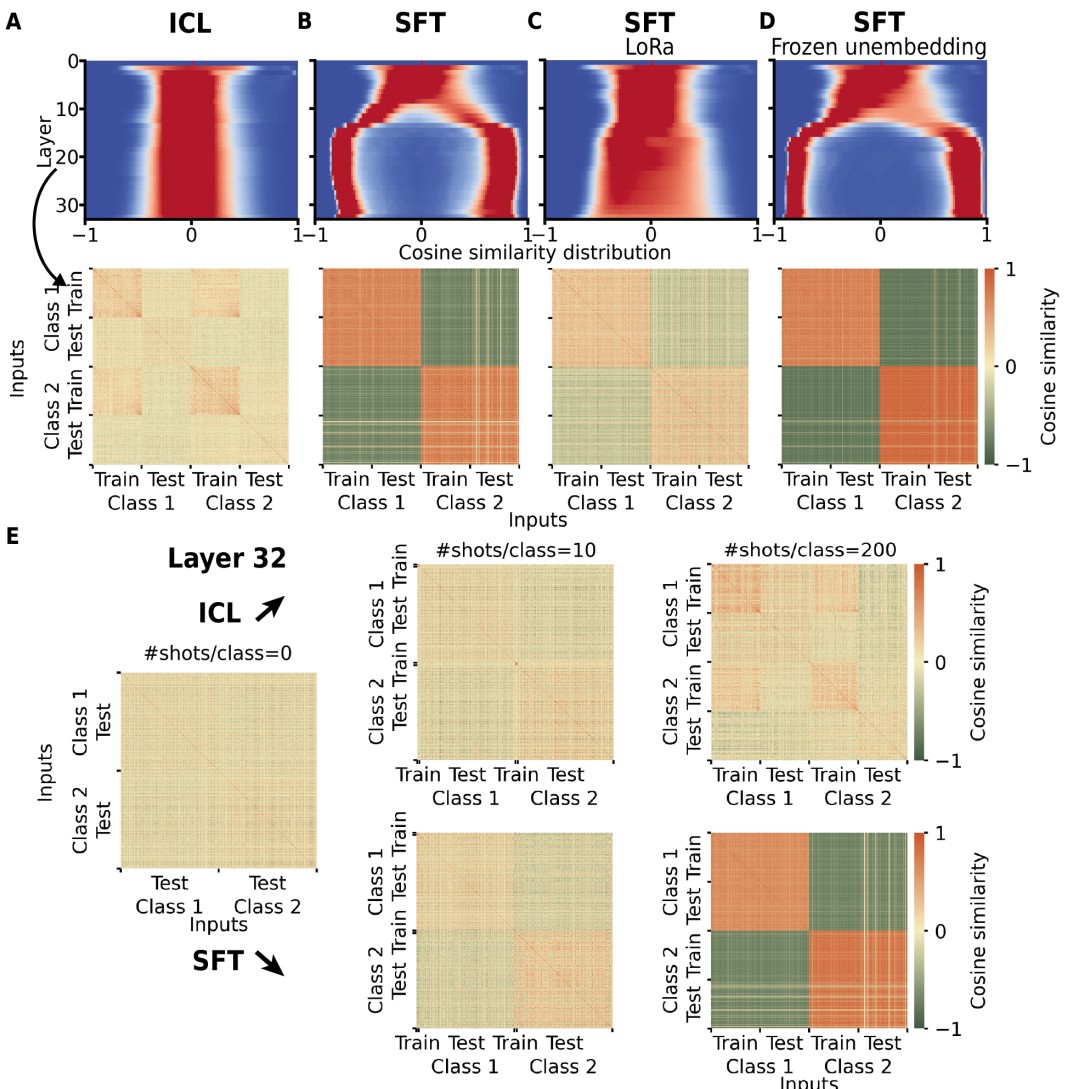

Figure 3: **Representations during ICL and SFT learning trajectories. A**: Top: RSA performed on the model activations at different layers. Cosine similarity was computed between last token activations on all 200 shots/class prompts ($10,000 \times 10,000$ matrix). For each transformer layer, the histogram of the RSA matrices is plotted. Bottom: Corresponding RSA matrix for layer 20, used to compute one row in the plot above. Only 400 inputs are compared, those part of the training set (in the context for ICL, or training examples for SFT), supplemented by randomly selected test inputs, sorted by labels. These plots were computed for Llama3-8B trained with ICL. **B**: Same as A, but for Llama3-8B trained with SFT (see Section A.3, "vanilla SFT") **C**: Same as A, but for Llama3-8B but finetuning using low-rank adaptation (LoRA, (Hu et al., 2022)). The analysis was otherwise identical, more in Section A.3. **D**: Same as A, but with the weights of the unembedding matrix frozen. The training method, hyperparameter values and analysis were otherwise identical. **E**: RSA on one model layer during an ICL and an SFT learning trajectory.

set elicited noticeably more similar representations, regardless of their class, than test examples or the same class (Fig. 3E). This was the case for all training examples, irrespective of their ordering in-context, both for 10 shots and 200 shots.

When analyzing the training dynamics of SFT for one fixed shot count (Supp. Fig. A7), it appeared that the observed collapse for SFT was tied to performance and not only a consequence of over-training, though the collapse appeared to increase with training, even after reaching a performance

plateau. Finetuning with low-rank adaptation (LoRA, (Hu et al., 2022)) mitigated the observed collapse, though the RSA matrix remained much more similar to SFT than to ICL (Fig. 3C). Finally, freezing the unembedding matrix during SFT did not eliminate the collapse, suggesting that said collapse was linked to task performance (Fig. 3D).

Overall, it appeared that despite reaching similar training and generalization performance on our controlled 2-D linear classification task, ICL and SFT did so with markedly different inductive biases and internal representations.

### 4.5 GENERALIZATION ACROSS MODELS AND TASK VARIANTS

Replaying an identical ICL or SFT trajectory across other LMs—Qwen3-8B (Yang et al., 2025) and Gemma3-12B/27B (Kamath et al., 2025)—revealed model-specific results (Fig. 4A,B,F,G), with several newer and larger models under-performing Llama-3-8B in terms of generalization performance and data efficiency. Nevertheless, the row/column and diagonal generalization patterns ("previously seen feature value bias" and "comparison bias", Fig. 1 & 2) were qualitatively conserved across models, especially in the few-shots case, indicating that the bias types reported above were not idiosyncratic to Llama3-8B (Fig. 4A,B,F,G). We also found representational collapse for SFT but not for ICL with Qwen3-8B (Fig. 4C,D). Once again, LoRA attenuated the collapse (Fig. 4E).

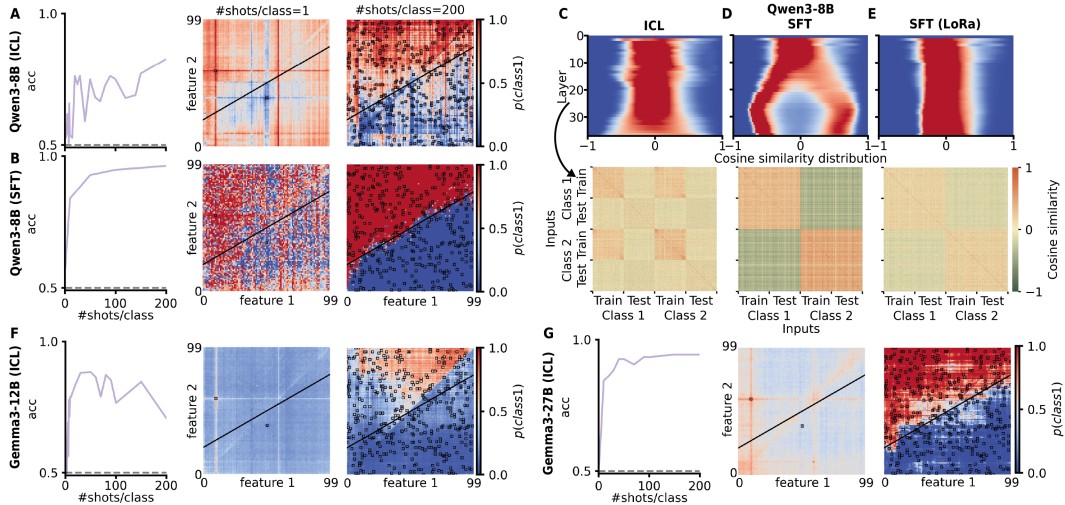

Figure 4: **Extension to other models.** **A,B**: Same trajectory with $\theta = 30°$ (exact same ordering of training examples) for Qwen3-8B, trained either with ICL (A) or SFT (B). From left to right: accuracy computed on all 10,000 possible inputs for the task as a function of the number of shots per class; visualization of the decision boundary of the model for increasing number of shots: probability associated with the logit of class 1 for all possible task inputs (same as in Fig. 1B). The probabilities are normalized for decision making such that $p(class\ 1) + p(class\ 2) = 1$. Black squares indicated the examples present in-context (ICL) or in the training set (SFT). **C-E**: Same visualization as Fig. 3A-C, but for Qwen3-8B; Top: RSA performed on the model activations at different layers. Cosine similarity was computed between last token activations on all 200 shots/class prompts ($10,000 \times 10,000$ matrix). For each transformer layer, the histogram of the RSA matrices is plotted. Bottom: Corresponding RSA matrix for layer 20, used to compute one row in the plot above. Only 400 inputs are compared, those part of the training set (in the context for ICL, or training examples for SFT), supplemented by randomly selected test inputs, sorted by labels. **F,G**: Gemma3-12B and Gemma3-27B trained with ICL on the same trajectory as A and B.

In addition, we devised a semantic version of the 2-D linear classification task by replacing integers with valence-ordered adjectives (e.g. Abysmal, Appalling, ... Subpar ... Decent ... Great ... Amazing, Fig. 5A). Performance still improved with shots, yet learning was overall much slower than in the numeric version (Fig. 5B,C and Supp. Fig. A4). Moreover, the comparison bias ($\theta = 45°$) and previously seen feature bias were present, albeit weaker (Fig. 5C). This suggested that lexical

priors interacted differently with the task geometry when the input manifold was semantic rather than numeric, but trends observed in the toy task overall held.

Finally, we trialed a non-linear version of the 2-D classification task, by performing an XOR operation on two linear tasks with $90°$ angle difference. Once again, we found that the trends from the linear task held, though learning was overall slower and task angles made less of a difference (Fig. 5E,F).

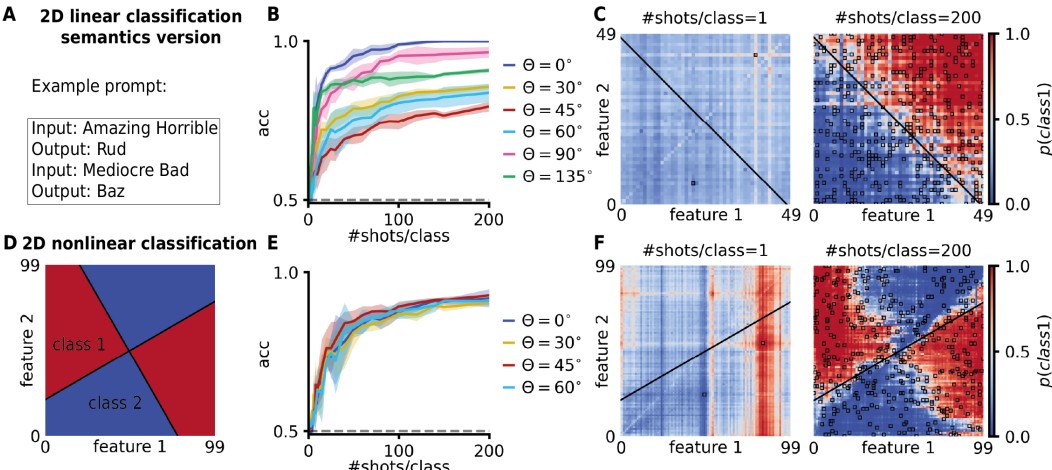

Figure 5: **Extension to other tasks. A**: Llama3-8B in a semantic version of the 2-D linear classification task, in which integers were swapped for adjectives ordered by valence. **B**: Accuracy during ICL computed over all 10,000 possible task inputs as a function of the number of shots per class (mean and standard deviation, computed over 10 example orderings), for different instances of the task (task angle $\theta$). **C**: Visualization of the decision boundary of the model for ICL for increasing number of shots: probability associated with the logit of class 1 for all possible task inputs (same as in Fig. 1B). The probabilities are normalized for decision making such that $p(class\ 1) + p(class\ 2) = 1$. Black squares indicated the examples present in-context. **D**: Non-linear 2D classification task, Llama3-8B: Visualization of ground truth labels of the task, red is class 1 and blue is class 2, for $\theta = 30°$. More details on the non-linear task in Section A.5. **E**: Evolution of the accuracy across shots/class for different task angles (mean and standard deviation, computed across 5 seeds per task angle) for ICL. **F**: Same as C for Llama3-8B ICL on the non-linear task.

## 5 DISCUSSION

In this work, we consider ICL and SFT as two learning algorithms and compare them on a controlled 2-D linear classification task with matched data and training example ordering. We observe that although the two strategies reach similar training and generalization performance, they do so with different inductive biases (Fig.1 & 2). In particular, ICL exhibits stronger priors on solving tasks likely inherited from pretraining, such as pattern-matching and number comparison, which affects its generalization patterns and performance. These biases on our synthetic task can be connected to documented behaviors in open-domain LMs, such as familiarity/retrieval—for instance, through induction heads (Olsson et al., 2022)— and magnitude-comparison heuristics (Srivastava et al., 2023; Nikankin et al., 2024; Shah et al., 2023). SFT is not unbiased either, though the motifs were a bit harder to pinpoint with our chosen task. That SFT has biases too is not a surprise, if anything because it acts on the same base model and its priors. Though we observe that these pretraining priors may be weakened or altered by SFT.

Conversely, at the level of the model's internal representations, ICL maintains richer, input-specific structure across layers, while SFT rapidly aligns internal states along label-separating directions (representational collapse), yielding higher confidence but reduced structural diversity (Fig.3 & 4). This was conserved across models and finetuning strategies (though specific methods such as LoRA appeared able to limit collapse), indicating that the observed representation collapse is a feature

of SFT and not an artifact of our model choice or the exact SFT strategy used. These differences of representations, also seen in other tasks (Doimo et al., 2024), may explain the observed angle-dependent accuracy and ordering susceptibility (Fig.2), and echo broader reports of SFT-induced specialization (and OOD fragility) (Lampinen et al., 2025; Chan et al., 2022b; Mosbach et al., 2023; Yin et al., 2024). We also predict that SFT may hinder transfer learning for similar reasons.

Our results are most consistent with ICL performing task-conditioned inference using priors from pretraining (consistent with Bayesian/implicit-optimizer views), rather than implementing literal gradient descent in medium-sized LMs. These findings challenge previous work in simplified settings (Von Oswald et al., 2023; Akyürek et al., 2022), and are in agreement with other reports in more realistic settings (Shen et al., 2023; Raventós et al., 2023).

The fact that inductive biases such as the comparison bias ($\theta = 45°$) were conserved across models and tasks during ICL (Fig. 4 & 5) suggests that at least parts of the ICL learning algorithm reflect more general natural language data properties rather than a model's specific architecture or idiosyncrasies of training. Such data properties have been shown to drive the emergence of ICL itself during pretraining (Chan et al., 2022a).

### Limitations

**Scope of tasks:** We focus on a single family of geometry-controlled 2-D classification tasks. While this isolates inductive biases and representations, it may not capture the complexities of hierarchical or multi-stage reasoning. Extending to more real-world in-context learning tasks would test the generality of our conclusions.

**Compute and model coverage:** The experiments were centered on medium-sized LMs. Scaling the ICL vs. SFT comparisons to larger models is an important next step.

**Hyperparameter breadth:** Though we performed several sweeps on SFT hyperparameters to investigate their influence on task performance (Supp. Fig. A2, A7, A8, A9, A10, A11 & A14), we cannot claim to have exhaustively investigated their influence on the model's inductive biases and representations. For instance, we have not exhaustively probed regularizers (e.g. weight decay schedules) or early-stopping/calibration strategies that could mitigate representational collapse.

**Prompt design and ordering controls:** Our ordering probes use synthetic periodic sequences. While they reveal strong short-horizon effects, broader prompt-engineering and selection strategies may alter the task-inference observed here, though systematically mapping this design space is beyond our scope.

**Representation readout:** RSA was performed on last-token activations. Alternative choices might reveal additional structure. Importantly, we only presented *correlational* evidence between differences in representations and differences in inductive biases, making causal manipulations an important follow-up.

ACKNOWLEDGMENTS

We would like to thank Andrew Lampinen, Daniel Wurgaft, Sara Dragutinovic and Yedi Zhang for useful discussions. This work was supported by a Schmidt Science Polymath Award, the Sainsbury Wellcome Centre Core Grant from Wellcome (219627/Z/19/Z) and the Gatsby Charitable Foundation (GAT3850).

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
