# A APPENDIX

## A.1 SUPPLEMENTARY FIGURES

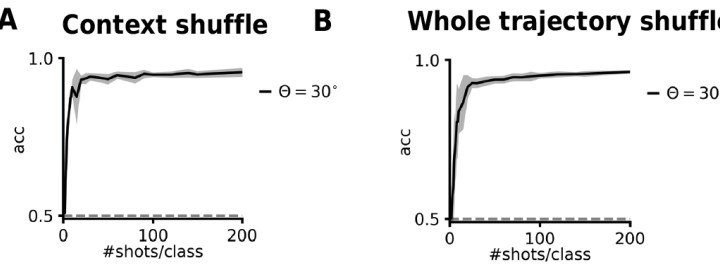

Figure A1: **Influence of the ordering of examples in-context A**: Llama3-8B on the $\theta = 30°$ task shown in Fig. 1, but the examples shown in context are shuffled (from the same pool of training examples from a fixed trajectory. **B**: Same as A, but the entire trajectory is changed across simulations (different training examples altogether, not only their ordering in-context).

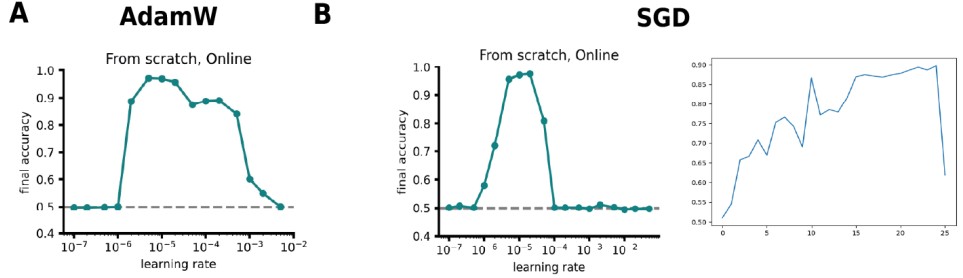

Figure A2: **Hyperparameter choices for SFT A**: Sweep of the learning rate (final held-out accuracy) for AdamW + cosine schedule. **B**: Left, same as A, but for vanilla SGD with constant learning rate. Right: example of an unstable SFT training with SGD: held-out accuracy as a function of the number of shots per class.

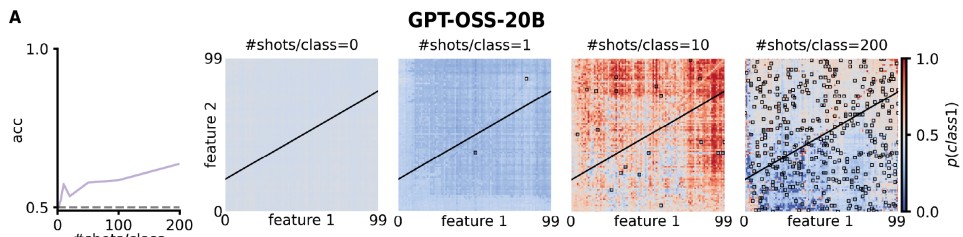

Figure A3: **Additional model comparison on an identical ICL trajectory. A**: Gpt-oss-20B on the $\theta = 30°$ task shown in Fig. 1 & 4. The plots obey the same structure as in Fig. 4A-D.

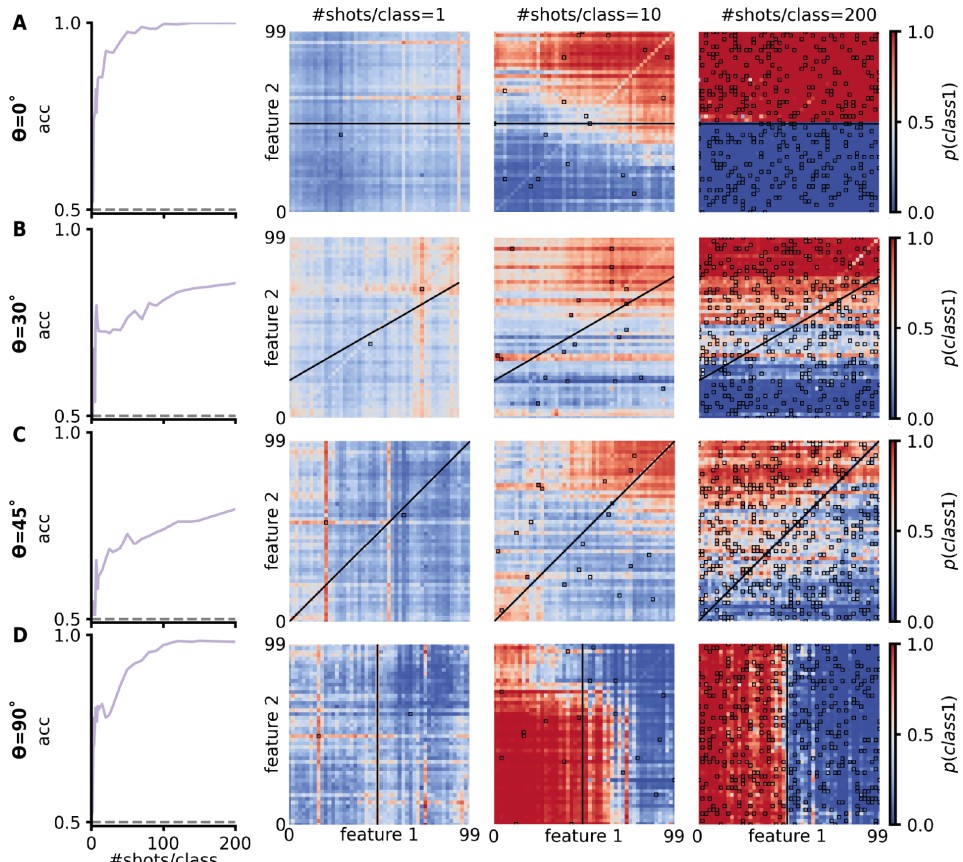

Figure A4: **Semantic 2-D linear classification. A-D**: From left to right: accuracy computed on all 10,000 possible inputs for the task as a function of the number of shots per class; visualization of the decision boundary of the model for increasing number of shots: probability associated with the logit of class 1 for all possible task inputs (same as in Fig. 1B). The probabilities are normalized for decision making such that $p(class\ 1) + p(class\ 2) = 1$.

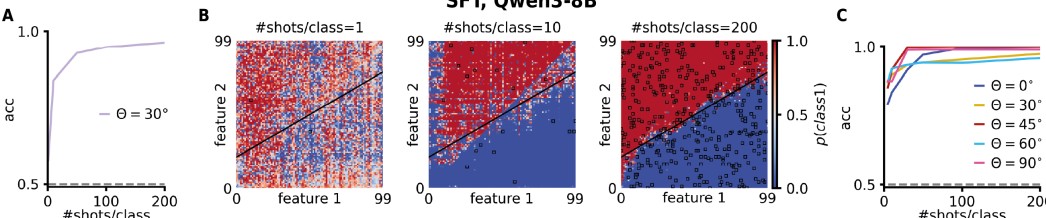

Figure A5: **SFT on Qwen3-8B. A**: Accuracy computed on all 10,000 possible inputs for the task as a function of the number of shots per class (generalization accuracy). **B**: Visualization of the decision boundary of the model for an increasing number of shots: probability associated with the logit of class 1 for all possible task inputs (same as in Fig. 1B). The probabilities are normalized for decision making, such that p(class 1) + p(class 2) = 1. **C**: Same as A for several task angles.

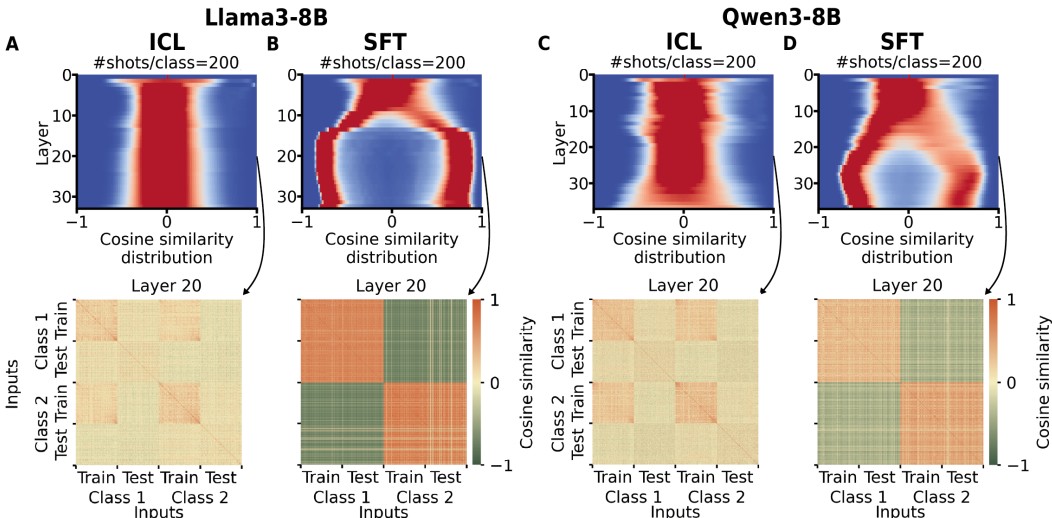

Figure A6: **Model representations during ICL and SFT. A**: Top: RSA performed on the model activations at different layers. Cosine similarity was computed between last token activations on all 200 shots/class prompts ($10,000 \times 10,000$ matrix). For each transformer layer, the histogram of the RSA matrices is plotted. Bottom: Corresponding RSA matrix for layer 20, used to compute one row in the plot above. These plots were computed for Llama3-8B trained via ICL. **B**: Same as A but for Llama3-8B trained with SFT ("Vanilla SFT", more in Section A.3). **C**: Same as A, but for Qwen3-8B. **D**: Same as B, but for Qwen3-8B.

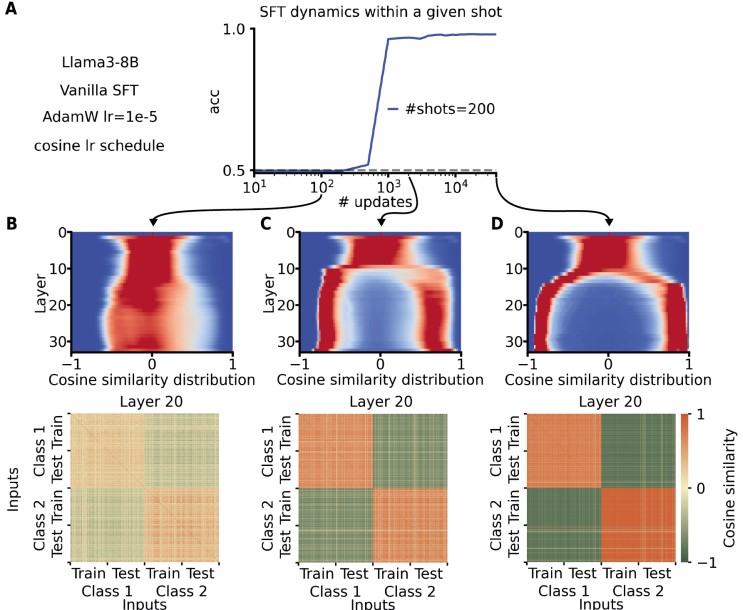

Figure A7: **Learning dynamics of vanilla SFT, Llama3-8B. A**: For a given number of shots (200), evolution of the generalization accuracy as a function of the number of updates to the model parameters (1 epoch is 400 updates for 200 shots). **B-D**: Same RSA analysis as in Supp. Fig. A6, but for model activations computed after 100 updates (B), 4k updates (C) and 40k updates (D).

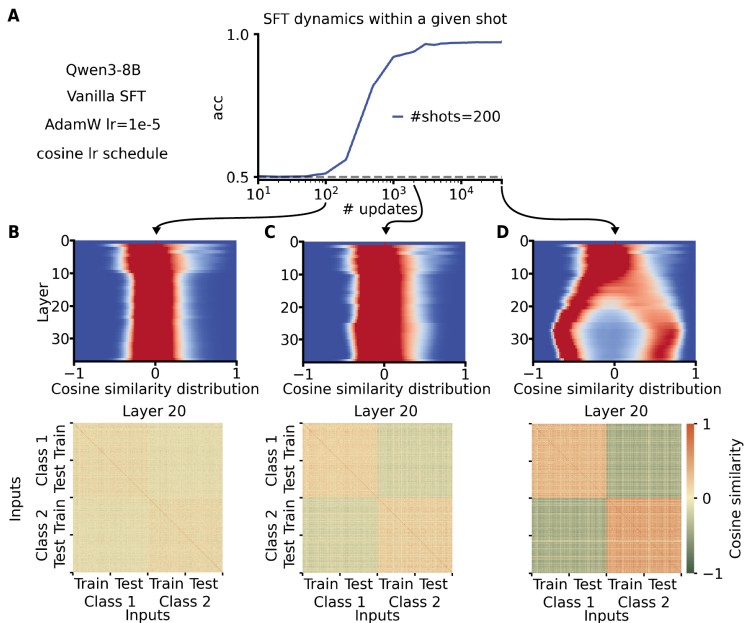

Figure A8: **Learning dynamics of vanilla SFT, Qwen3-8B**: Same as Supp. Fig. A7, but for Qwen3-8B. The training method, hyperparameter values and analysis were otherwise identical.

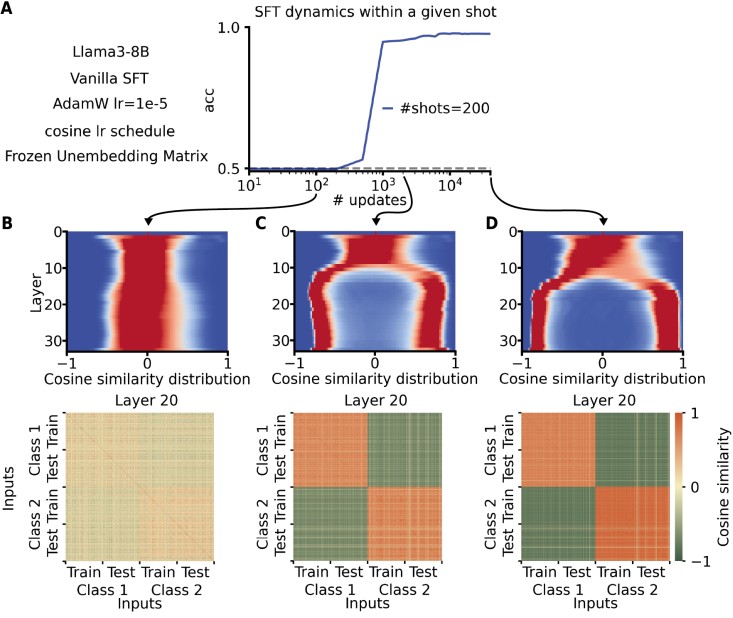

Figure A9: **Learning dynamics of vanilla SFT with frozen unembedding matrix, Llama3-8B**: Same as Supp. Fig. A7, but with the weights of the unembedding matrix frozen. The training method, hyperparameter values and analysis were otherwise identical.

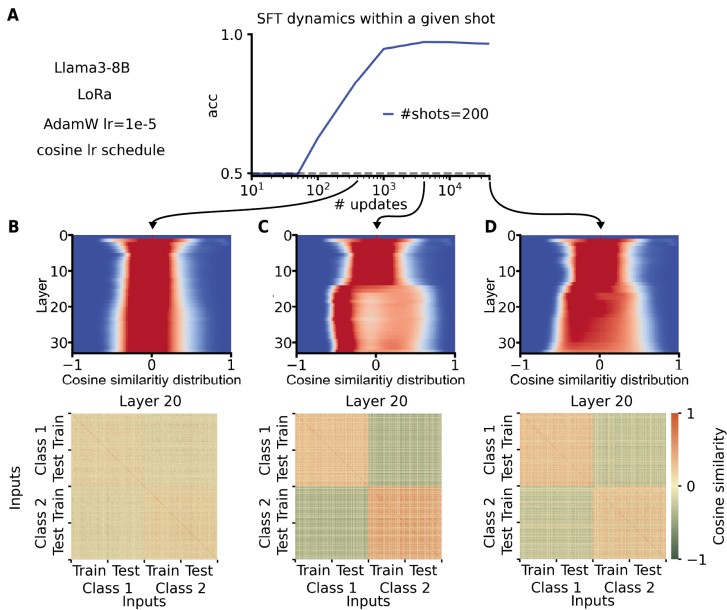

Figure A10: **Learning dynamics of LoRA, Llama3-8B**: Same as Supp. Fig. A7, but training using low-rank adaptation (LoRA, (Hu et al., 2022)). The analysis was otherwise identical, more in Section A.3.

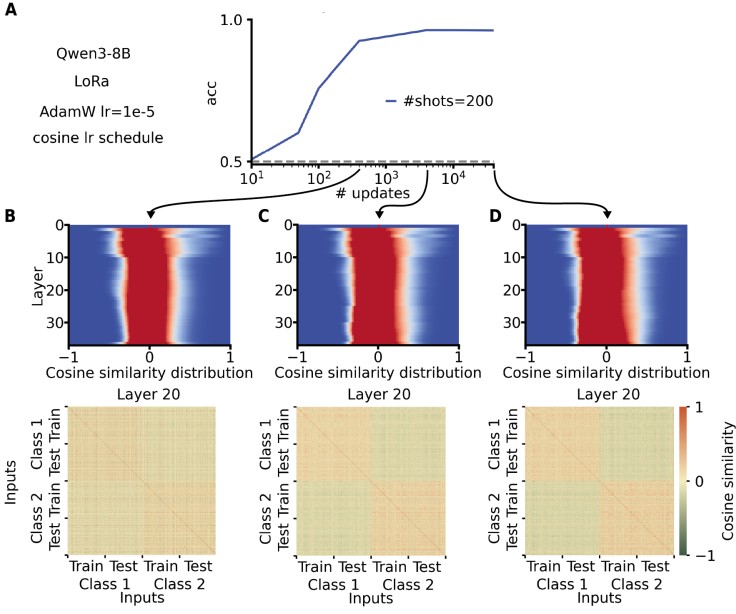

Figure A11: **Learning dynamics of LoRA, Qwen3-8B**: Same as Supp. Fig. A10, but for Qwen3-8B. The training method, hyperparameter values and analysis were otherwise identical.

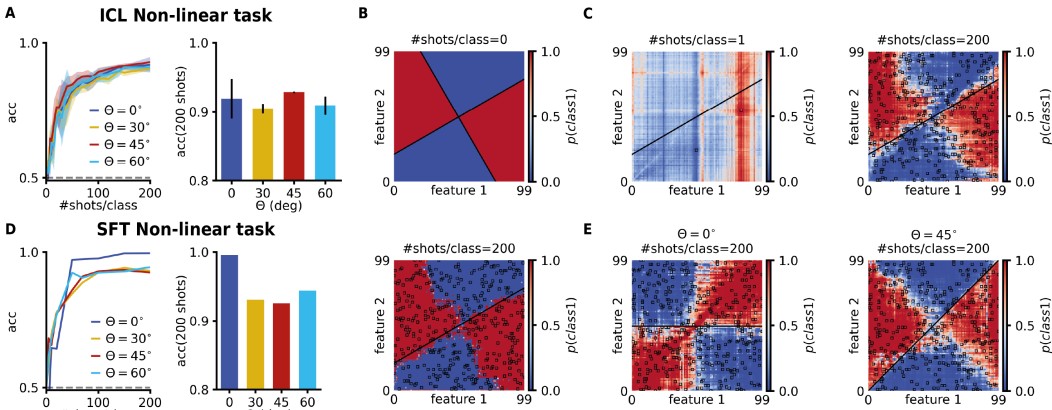

Figure A12: **Non-linear 2D classification task, Llama3-8B. A**: Left: evolution of the accuracy across shots/class for different task angles (mean and standard deviation, computed across 5 seeds per task angle) for ICL. Right: accuracy for 200 shots/class (mean and standard deviation, computed across 5 seeds per task angle) for different task angles. **B**: Visualization of ground truth labels of the task, red is class 1 and blue is class 2, for $\theta = 30°$. More details on the non-linear task in Section A.5. **C**: Example trajectories for ICL, for 1 and 200 shots per class, the probability associated with the logit of class 1 for all $100 \times 100$ possible inputs in the task. The probabilities are normalized for decision making such that $p(class\,1) + p(class\,2) = 1$. The black line denotes the ground-truth decision boundary ($\theta = 30°$). Black squares indicated the examples present in-context (ICL) or in the training set (SFT). **D**: Left, middle: Same as A, but training with vanilla SFT instead. Right: Same as C for 200 shots. **E**: Visualization of model generalization behaviour like in C and D, for ICL and other task angles.

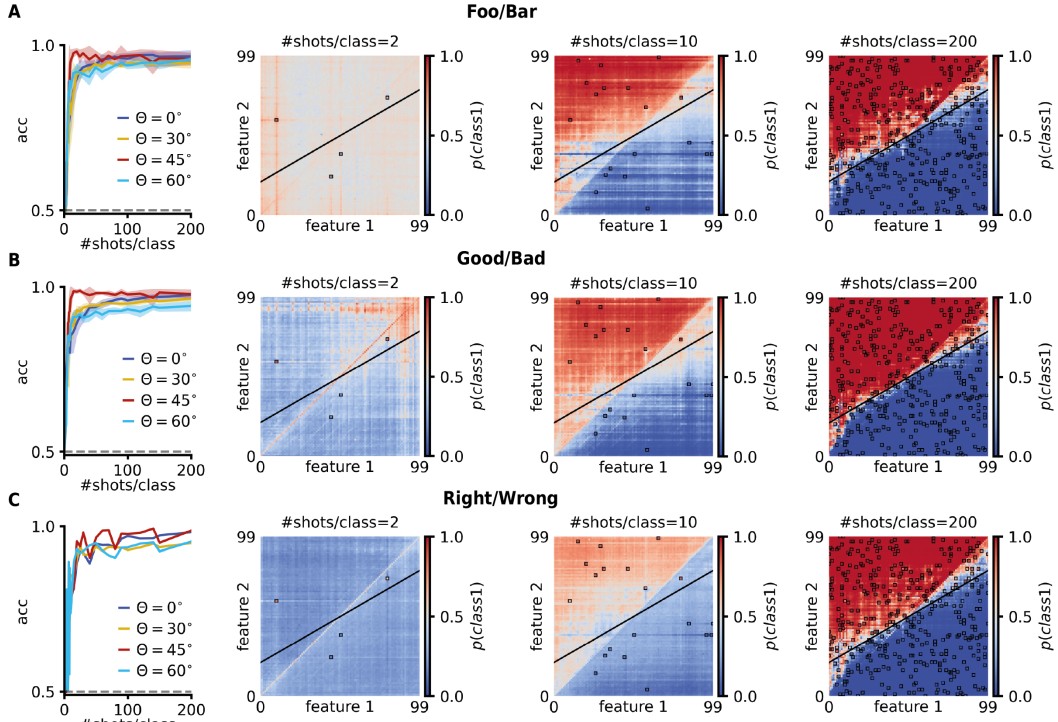

Figure A13: **Other class labels, Llama3-8B**: Same trajectories for several task angles for Llama3-8B on the linear 2D classification task, but with different label pairs: " Foo"/" Bar", " Good"/" Bad" and " Right"/" Wrong". We verified that all were one-token long. **A-C**: From left to right: accuracy computed on all 10,000 possible inputs for the task as a function of the number of shots per class; visualization of the decision boundary of the model for increasing number of shots: probability associated with the logit of class 1 for all possible task inputs (same as in Fig. 1B). The probabilities are normalized for decision making such that p(class 1) + p(class 2) = 1. Black squares indicated the examples present in-context (ICL) or in the training set (SFT).

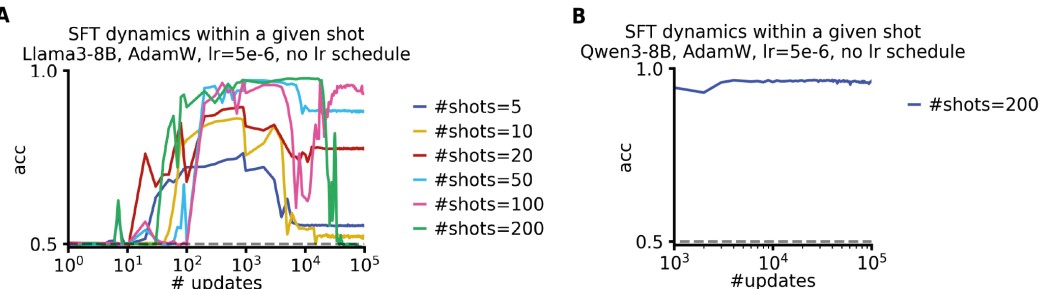

Figure A14: **SFT hyperparameters for SFT. A**: Generalization accuracy of Llama3-8B with Vanilla SFT *without* the cosine learning rate schedule, as a function of the number of updates to the model. Plot repeated for different numbers of shots (i.e. different numbers of unique training samples). **B**: Same as A for Qwen3-8B and 200 shots.

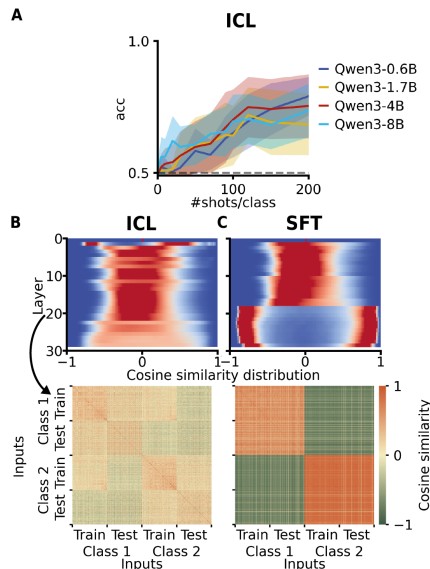

Figure A15: **Smaller models. A**: Generalization accuracy of different models from the Qwen3 family as a function of the number of shots on matched trajectories from the $30°$ linear task. Means and standard deviations computed over 15 seeds. **B,C**: Same as Fig. 3A&B, but for Qwen3-0.6B, 200 shots, $30°$ linear task.

## A.2 SEMANTICALLY UNRELATED LABELS

The typically used "Foo" and "Bar" labels for semantically unrelated ICL already have a few years of existence in the literature (Min et al., 2022; Wei et al., 2023), so it can be assumed that they are part of the pre-training set somehow. Moreover, "Bar"—unlike "Foo"—is an English word which creates a bias towards "Bar", at least in the few-shot regime. Since our work studies the unfolding of the learning dynamics of ICL, we chose two other sequences of letters with single token representations for most open-source tokenizers and more balanced default priors. Note that we tested the influence of the label pair on the ICL learning dynamics in Supp. Fig. A13.

## A.3 SFT TRAINING DETAILS

### A.3.1 "VANILLA SFT"

We trained all the parameters of the model using PyTorch with the AdamW optimizer and a cosine learning rate schedule. After parameter sweeps (Supp. Fig. A14 and A2), we chose a learning rate of $10^{-5}$, 100 epochs, a warmup ratio of 0.05 and final learning rate 1e-7 for the learning rate schedule. This was chosen once for all LMs tried. We initially favored a simpler learning rule (vanilla SGD), but this led to some optimizations unexpectedly blowing up and making results inconsistent (Supp. Fig. A2). The learning rate schedule was also instrumental in stabilizing training (Supp. Fig. A14).

### A.3.2 LORA

For Supp. Fig. A10 and A11, we used low-rank adaptation (LoRA) from the peft library from HuggingFace, with matrices of rank 8.

## A.4 SEMANTIC VERSION OF THE 2-D LINEAR CLASSIFICATION TASK

We use a list of 50 adjectives of increasing valence as a replacement for integers in an otherwise identical classification task with the same single token output tokens described above.

### A.5 NON-LINEAR 2-D TASK

This version of the task had exactly the same inputs as for the linear task. For a given task angle $\theta \in [0, 90]^\circ$, the labels were computed by doing XOR of the labels from the linear task with angle $\theta$ and $\theta + 90^\circ$ (See Supp. Fig. A12B for example with $\theta = 30^\circ$).

### A.6 NOTE ON ADDITIONAL MODELS

The poor results on gpt-oss-20b (Agarwal et al., 2025) are likely due to the fact that this model explicitly expects OpenAI's harmony format for chat (i.e. is not purely a base model).