# OpenReview forum: "Comparing the learning dynamics of in-context learning and fine-tuning in language models"
_ICLR.cc/2026/Conference — ICLR 2026 Poster_

### Official Review · Reviewer_H2hK · 2025-10-27

**Soundness:** 2
**Presentation:** 2
**Contribution:** 3
**Rating:** 4
**Confidence:** 4

**Summary:**

The authors apply both few-shot prompting ("ICL") and supervised fine-tuning ("SFT") to a toy 2D linear classification task, analyzing how performance and representations change as more examples are added.

**Strengths:**

* Demonstrates that while final performance is similar between ICL and SFT, (a) ICL has a prior towards 45 degrees while SFT has a prior towards 90 degrees, (b) ICL is sensitive to periodic orderings of example labels, and (c) SFT representations are more tightly coupled to the label space.
* Finds support for (a) across Llama3-8B, Qwen3-8B, Gemma3-12B, and Gemma3-27B

**Weaknesses:**

* Claims in the text are often not clearly linked to the corresponding portions of the dense and complex figures. For example, the claim "Both predictions were verified when comparing model performance across seeds for ICL" references "Fig.2A" when I believe the only relevant part of Figure 2A is column 4, the claim "we observed an overestimation (resp. underestimation) of the inferred task angle for θ = 30◦ (resp. θ = 60◦)"  references "Fig.2A" when I believe the only relevant part of Figure 2A is column 3, etc. Minimally, every subfigure within the current figures needs to be labeled somehow (not just the rows), and the appropriate subfigure needs to be referenced by label wherever it is discussed. More broadly, every claim should be coupled with text that explains how to read that claim off of the corresponding figure.
* The focus is on a single artificial task, and how these findings might generalize to other problems is not well discussed. For example, I don't know what prediction the identified bias towards a 45 degree angle would make for a real-world task like question answering.
* Most analyses are done on Llama3-8B, so we can't tell, for example, whether findings (b) and (c) listed in the Strengths generalize to other LLMs. (The appendix shows that GPT-OSS:20B can't even learn the task, so generalization of the findings across models is an important concern.)

**Questions:**

* I don't understand how to read the "previously seen feature bias" off the figures. Can you explain?

---

> ### Comment · Reviewer_H2hK · 2025-11-24
>
> Thanks for the detailed response. The Qwen results you added substantially strengthen the (c) claim. I have raised my Rating.

---

> > ### Author Response · Authors · 2025-11-28
> >
> > We are very happy that the additional experiments performed have improved your assessment of our submission. For completion, besides the points raised in the common rebuttal, we summarize below the changes made in the paper following your original comments:
> >
> > ### 1/ Linking claims in the text to precise parts of the figure:
> > We agree with this point and have done our best to correct it in the revised version of the paper.
> >
> > ### 2/ Failure of GPT OSS:
> > The failure of this model to learn the task is because it is an instruct-tuned model and is expecting a specific formatting of inputs and outputs, which collides with our minimal controlled prompts. We only report it for the sake of completion, but we have added a warning in the revised version and we don’t think this puts the other results of this work into question, especially given that we’ve extended our all key results in multiple model families and sizes (Llama, Gemma, Qwen, 0.6B-27B)
> >
> > ### 3/ Previously-seen feature bias:
> >
> > We observed that for 1-shot in ICL (Fig.1B second column, ICL; Fig.4A,B,F,G second column) the model tended to attribute the same label to inputs sharing one feature value with the training sample (the dark red and dark blue rows and columns). Sometimes, we also see that inputs with feature 2 being the same as training feature 1, which creates another row and column pattern related to the training sample.

---

### Official Review · Reviewer_sBMX · 2025-11-01

**Soundness:** 3
**Presentation:** 3
**Contribution:** 3
**Rating:** 6
**Confidence:** 4

**Summary:**

The paper directly contrasts in‑context learning (ICL) and supervised fine‑tuning (SFT) as learning algorithms on a tightly controlled 2‑D linear classification family. Using matched data, shot counts, and example orderings, the authors track accuracy, "smoothness," confidence, inferred boundary angle, and layer‑wise representational similarity (RSA). Both ICL and SFT reach similar held‑out accuracy, but with different inductive biases and internal representations: ICL preserves richer input structure yet shows stronger from-pretraining priors (e.g., diagonal "number‑comparison" bias and row/column reuse), whereas SFT aligns representations along label axes, yielding higher confidence but apparent representational collapse. Ordering effects reveal short‑horizon pattern‑following in ICL. Results qualitatively persist across several model families and in a semantic (adjective‑ordered) variant, though learning there is slower.

**Strengths:**

* The matched‑trajectory setup isolates algorithmic differences between ICL and SFT more cleanly than typical open‑domain benchmarks

* I think the comparison and framing of SFT and ICL as two distinct learning algorithms is a useful and interesting framing

**Weaknesses:**

* I think the main limitation, as mentioned by the authors, is scope: Add one or two richer tasks (non‑linear boundaries or 3‑class variants; a small real‑text task with controlled geometry) to test whether the same ICL/SFT differences recur.

* RSA uses last‑query‑token activations; alternative readouts (earlier tokens, attention heads, probing classifiers) could nuance the "collapse vs. preservation" story. Causality is not established

* Fig. 3 compellingly shows SFT’s label‑aligned compression vs. ICL’s input‑structured geometry, even when accuracies match. I thought this was cool

**Questions:**

Does representational collapse happen with LoRA or other PEFT approaches?

---

> ### Author Response · Authors · 2025-11-28
>
> We thank you for the valuable comments and positive evaluation of our work. We have summarized the additional experiments performed in the common response. Please find below more details on the specific points raised by your review.
>
> ### 1/ Scope of tasks:
>
> We think that our simple task is both a strength and a limitation of this work, as it enables this “matched-trajectory setup” praised by the reviewer, as well as a simple analysis and visualization of a model’s inductive biases and generalization behaviour.
> We have extended the linear 2D task to a semantic version (same structure, but instead of numerical inputs, we have adjectives of increasing valence, see Fig.5). Now we have also devised a non-linear version, which is essentially an XOR of two linear tasks, and find similar results than on the linear task.
>
> ### 2/ Collapse under LoRa:
>
> We tried LoRa on Llama3-8B (Supp. Fig. 15) and Qwen3-8B (Supp. Fig. 16). Your intuition was right, LoRa does seem to mitigate the collapse. We have added this to the main paper and the discussion. Though, despite reducing the collapse observed, the corresponding RSA matrices remain more similar to other SFT runs rather than ICL.
> We have tried other PEFT approaches such as prompt-tuning, but we are not able to reach good performance on the task as of yet.

---

### Official Review · Reviewer_fv6R · 2025-11-01

**Soundness:** 3
**Presentation:** 4
**Contribution:** 3
**Rating:** 6
**Confidence:** 4

**Summary:**

This work empirically compares the learning dynamics of in-context learning and finetuning on mid-sized language models on two synthetic tasks: a 2-D numerical classification task and a 2-D semantic classification task. This work demonstrates that, while both algorithms can learn the tasks, in-context learning is more subject to biases present in natural language training data (e.g., biases towards comparing numbers, rather than learning an arbitrary classification boundary). Furthermore, the authors demonstrate that the two learning algorithms yield radically different internal representations, with finetuning collapsing representations into two classes and ICL maintaining a greater amount of structure in representations.

**Strengths:**

This work empirically addresses a broad, pressing question in the field: what is the empirical relationship between in context learning and finetuning?

The work appears empirically sound, the experiments are thorough, and the breadth of models tested (within the mid-sized scale) is fairly large.

The results regarding inductive biases of ICL are sensible, yet interesting. These results bolster broader arguments that ICL is selecting from a pool of functions learned in pretraining. The analyses of inductive bias in ICL add nuance to this discussion, indicating that there are systematic differences in the prior probability assigned to functions.

**Weaknesses:**

Why not study small models as well? Even if studying very large models is computationally challenging, one might be able to discover a scaling trend in ICL vs finetuning learning dynamics by systematically studying models on the order of 1B to 27B. This would increase the impact of this work.

The RSA of the finetuned models should be further fleshed out. One guess is that the task is so trivial that the model simply converges on the correct answer early on in its computation (i.e., at an early layer). Perhaps early on in finetuning, the model’s representations look more like what you find for ICL, or perhaps there is a one-to-many comparison to be made between the early layers of the finetuned model and all of the layers in the ICL model. This setup seems especially well suited for a finding like “finetuning results in a compressed version of the computation that ICL converges on”. Studying the dynamics of what is called “representation collapse” in the text would be a very valuable contribution.

**Questions:**

Why not study smaller models as well?

Did you try using other nonsense labels? Or even labels with pretrained semantics? Perhaps there would be interesting divergences between ICL and finetuning that hinge on the label.

---

> ### Author Response · Authors · 2025-11-28
>
> We thank the reviewer for their encouraging comments and interesting suggestions. We refer the reviewer to our common response to all reviewers detailing additional experiments. In addition, please find below a more detailed answer to the points raised in this review.
>
> ### 1/ Why not study smaller models as well?
>
> We took advantage of the Qwen3 series which has an ideal range of sizes (0.6B, 1.7B, 4B and 8B), and compared their performance in ICL, and their representations for ICL vs SFT (Supp. Fig. 20). We did not find striking differences in performances, generalization patterns or representations. We believe there probably is something to understand and analyze there, but it will require novel analyses that fall beyond the scope of the current work.
>
> ### 2/ Fleshing out the RSA analysis:
>
> We believe that the extensive additional analysis of finetuning and collapse across models and methods presented in the common response addresses this concern.
> In particular, the learning dynamics plots (e.g. Supp. Fig. 12) directly tackle the “dynamics of representational collapse” point: we show that even early in finetuning, the representations are different to ICL, which is verified across models and training strategies (e.g. Supp. Fig. 13-16). We also think that the histogram of rsa plot (e.g. Supp. Fig. 11 top row) nicely shows the dynamics of collapse across layers. Moreover, the frozen unembedding experiment (Supp. Fig. 14) shows that meaningful parameter updates during SFT are not constrained to the unembedding matrix. We found there are many other interesting analyses that could be done, and we believe that this will be exciting future work.
> We thank the reviewer for prompting us to investigate this matter further.
>
> ### 3/ Nonsense labels:
> We tried 3 other label pairs (Supp. Fig. 19, point 4/ in common response), and did not find meaningful differences across label pairs.

---

### Official Review · Reviewer_b4Qy · 2025-11-03

**Soundness:** 3
**Presentation:** 4
**Contribution:** 2
**Rating:** 6
**Confidence:** 4

**Summary:**

This paper investigates the different inductive biases and learning dynamics of in-context learning (ICL) and supervised fine-tuning (SFT) in medium-sized language models. Using a controlled 2D linear classification task, the authors directly compare ICL and SFT across matched learning trajectories (i.e., same data and example ordering), analyzing both generalization patterns and internal representations. The authors find that while both methods achieve similar final accuracy, their strategies and inductive biases differ significantly. ICL is shown to exhibit strong inductive biases inherited from pretraining, such as a "comparison bias" (favoring diagonal decision boundaries, $\theta \approx 45^\circ$) and a "previously-seen feature value bias" (favoring axis-aligned boundaries). In contrast, SFT is shown to suppress task-irrelevant features, leading to internal states clustering primarily by task label. Conversely, ICL preserves more varied input-specific representations throughout its layers. These core findings are also shown to generalize to an analogous semantic classification task.

**Strengths:**

1. *Clear Presentation:* The paper does an excellent job of clearly presenting its results. The use of a minimal, controlled 2D linear classification task allows for a precise and direct comparison of the learning dynamics, generalization patterns, and inductive biases of ICL and SFT.
2. *Representational Analysis:* The representational similarity analysis (RSA) provides a clear distinction between the two learning regimes. It visually and quantitatively demonstrates SFT's representation collapse versus ICL's preservation of input structure (Fig. 3), adding a strong layer of evidence beyond simple task performance metrics.
3. *Strong Grounding in Literature:* The findings are well-situated within the broader literature, particularly in relation to Bayesian accounts of ICL, and contribute compelling evidence to the ongoing discussion challenging "ICL as gradient descent" mechanisms in larger models.

**Weaknesses:**

1. *Limited Scope of SFT Experiments:* The paper's central claims about SFT rely on fine-tuning experiments conducted primarily on a single model (Llama-3-8B), which the authors acknowledge as a limitation. While the ICL results are replicated across several models, the core ICL vs. SFT comparison would be greatly strengthened by a more comprehensive evaluation of the SFT condition (e.g., across more models, training hyperparameters, or regularizers) to ensure the "representation collapse" is a general feature of SFT and not an artifact of a specific setup.
2. *Novelty in Context of Prior Work:* While the direct comparison between ICL and SFT is valuable, many of the core results (e.g., ICL can be sensitive to pretraining priors, SFT can be brittle, etc) have been documented in related work. The paper could do a clearer job of articulating the specific, novel contribution of its findings beyond these effects. For instance, is the key novelty the direct demonstration of how the internal representations diverge under identical data, or the specific characterization of the "comparison" vs. "axis-aligned" biases?

**Questions:**

1. *Influence of SFT Hyperparameters*: The authors note that SFT hyperparameters were not exhaustively probed (Limitations, p. 9). Given that SFT is known to be sensitive to hyperparameter choices, how confident are the authors that the observed "representation collapse" is an inherent feature of SFT on this task, rather than an artifact of a specific hyperparameter regime (e.g., potential over-fitting)?
2. *Clarifying Novel Contribution*: The paper's discussion notes that ICL's sensitivity on pretraining priors and SFT's brittleness have been noted in prior work. Is the authors primary contribution the direct, matched-data demonstration of how ICL and SFT strategies diverge representationally, or the specific characterization of the competing inductive biases (e.g., "comparison bias" vs. "feature value bias")? A clearer framing of the novelty would help situate the work's impact.

---

> ### Author Response · Authors · 2025-11-28
>
> We thank the reviewer for their positive appraisal of our work and helpful comments. We refer the reviewer to our common response to all reviewers detailing additional experiments. In addition, please find below a more detailed answer to the points raised in this review.
>
> ### 1/ Limited scope of SFT experiments/Influence of SFT parameters:
> - We performed SFT runs with Qwen3-8B, along with a new visualization of representational collapse, which revealed qualitatively similar results to Llama3-8B (Supp. Fig. 10&11).
> - We unpack the dynamics of SFT during training within a single epoch, for Llama3-8B (Supp. Fig. 12) and Qwen3-8B (Supp. Fig. 13)
> We tried freezing the unembedding matrix to make sure the changes in intermediate layers in which the collapse is observed are important for learning (Supp. Fig. 14).
> - We tried another fine-tuning strategy, LoRa, both in Llama3-8B (Supp. Fig. 15) and in Qwen3-8B (Supp. Fig. 16). Despite reducing the collapse observed, the corresponding RSA matrices remain more similar to other SFT rather than ICL.
> - We report the effect of the learning rate (Supp. Fig. 7A), optimizer  (Supp. Fig. 7B) and cosine learning rate schedule (Supp. Fig. 19).
>
>
> Overall, within our compute budget and the scope of our work, we strengthened our SFT results. We are now confident that the collapse is a feature of SFT on this task and not an artifact of our model/hyperparameter choice.
>
> ### 2/ Novelty in the context of prior work/clarifying novel contribution:
>
> We agree that our paper makes several disparate novelty claims. While we believe it stems from an attempt to fairly acknowledge all neighbouring contributions present in the literature, we have tried clarifying and unifying these in the revised version. We also believe that with our now stronger results on SFT collapse, the contribution of the paper is clearer.

---

### Author Response · Authors · 2025-11-24
**Response to all reviewers with a summary of the additional experiments performed**

We thank the reviewers for their overall positive appraisal of our work, praising its presentation, “an excellent job of clearly presenting its results” (Reviewer b4Qy), “well-situated within the broader literature” (Reviewer b4Qy). Our comparison of ICL and SFT was deemed a “useful and interesting framing” (Reviewer sBMX), addressing “a broad, pressing question in the field” (Reviewer fv6R). Our 2D linear task enabled a “matched‑trajectory setup [that] isolates algorithmic differences between ICL and SFT more cleanly than typical open‑domain benchmarks” (Reviewer sBMX), while the RSA analysis “add[s] a strong layer of evidence beyond simple task performance metrics” (Reviewer b4Qy), in a “compelling” and “cool” way (Reviewer sBMX). The breadth of the models tested was also appreciated (Reviewer H2hK & fv6R).

Nevertheless, the questions raised were very helpful, and triggered many additional experiments on our side, which we believe have substantially improved our contribution. We summarize the main additions below. We will incorporate them in the main text and figures over the coming days, **for now they are in the appendix (Supp. Figs 10-19)**.

### 1/ Extending the SFT results to other models (Reviewer b4Qy):

We performed **SFT on Qwen3-8B**, with qualitatively similar findings to Llama3-8B (Supp. Fig.10). The extension to Gemma3-12B and 27B is underway, though computationally more arduous given our compute resources.

### 2/ Extending RSA + SFT results: dynamics of representational collapse across layers and training (Reviewers b4Qy, fv6R):

- We introduce a **novel visualization to probe for representational collapse with RSA**, by showing histograms of the flattened RSA matrices across layers (Supp. Fig.11A&B, Llama3-8B) to visualize the evolution of representations more easily. We see that the rsa matrices with SFT do become bimodal, indicating collapse.
Likewise, with Qwen3-8B (Supp. Fig.11C,D), we show evidence of representational collapse for SFT, but not for ICL. We also note that the RSA matrices for ICL look qualitatively similar across Llama3-8B and Qwen3-8B, and likewise for SFT.

- We plot the **RSA dynamics during SFT training within a number of shots** (Reviewer fv6R, Supp. Fig.12, Llama3-8B). Even early in fine-tuning, right after performance starts plateauing (Supp. Fig.12C), the RSA matrix already collapses. We reproduced this analysis for Qwen3-8B (Supp. Fig.13). In that case, early after reaching the performance plateau, the representations were still clustered by labels (Supp. Fig.13C bottom, to compare with Supp. Fig.11C bottom), though the collapse was less pronounced than for Llama. Thus there are model differences in how and when representations collapse with SFT. But we still see clear differences between the representations between SFT and ICL.
We note, as Reviewer fv6R predicted, the more we train the model after performance plateaus, the more the representation collapses  (Supp. Fig.12&13), though without impairing generalization performance (no overfitting).

- **SFT with frozen unembedding matrix**: we were curious how much of the collapse was due to unembedding updates rather than a meaningful change of the features throughout the layers. We observed a collapse even when this matrix was frozen during training. Though it remains correlational, it strengthens our hypothesis that the collapse observed is directly related to learning the task with SFT (Supp. Fig.14).


### 3/ Other SFT strategies (Reviewer sBMX):

- **LoRa**: We tested low-rank adaptation which also led to near perfect generalization performance (Supp. Fig.15). Interestingly, we found also evidence of representation collapse (Supp. Fig.15), though not as strong as full-rank SFT. For Qwen3-8B, the collapse could not be seen in the histogram plots, only the RSA plots show a clustering by label, like for the other SFT trajectories. Therefore, as suggested by reviewer sBMX, different finetuning strategies appear to affect models’ internal representations differently, though we could still tell apart SFT from ICL in all cases we tried.

- **Other PEFT approaches**: this is ongoing, but we would like to probe other strategies such as prefix tuning.

Taken together, the analyses above make us more confident that representations do differ between ICL and SFT. In particular, representational collapse appears to be a feature of SFT compared to ICL and not an artifact of our hyperparameter choices or chosen, though some fine-tuning methods used by practitioners such as LoRa may help mitigate this collapse. We thank the reviewers for their help in addressing this weakness in the original submission.

Edit: fixed figure numbers following upload of revised manuscript

---

> ### Author Response · Authors · 2025-11-24
>
> ### 4/ Other tasks (Reviewer sBMX):
>
> We extended our set-up to a **non-linear version of 2D classification**, akin to an XOR problem (Supp. Fig.17). We verified that our findings extended to this task for ICL (Supp. Fig.17A,C,E) and SFT (Supp. Fig.17D). Taken together with the semantic version of the linear task (Fig.5), we feel that we have now more exhaustively explored interesting tasks that still fall within the scope of this project, focusing on controllable tasks with easy analysis and understanding of generalization patterns and inductive biases.
>
> ### 5/ Other labels (Reviewer fv6R):
>
> We ran ICL experiments on Llama3-8B using 3 other output label pairs (Supp. Fig.18): another semantically unrelated pair (“Foo”/”Bar”), as well as two semantically-heavy label pairs (“Good”/”Bad” and “Right”/”Wrong” ). We find qualitatively similar results to the label pair used in the paper (“Rud”/’Baz”). We note, however, that for the first few shots, a strong preference of the model towards one of the two labels may obscure the generalization biases that appear clearly with the semantically unrelated label choice.
>
> ### 6/ Smaller models (Reviewer fv6R):
>
> We agree with this suggestion and are still in the process of running experiments. We will report shortly on this.
>
> ### 7/ More on SFT parameters and learning dynamics:
>
> In the paper, we performed SFT with online training with the AdamW optimizer, a fixed number of epochs and a cosine learning rate schedule (added section Supp.A.3). Following Reviewer b4Qy, we investigated in more detail the training dynamics of single SFT runs (as a function of the number of training samples for a fixed shot). We show sweeps on the learning rate (Supp. Fig.7&19). We show that removing the learning rate schedule destabilized training, in a model-dependent way (Qwen3-8B appeared more robust to this change than Llama3-8B).
>
> Overall, we believe we have addressed the major points raised in the reviews. We have some experiments left underway, for which we will report as soon as possible (1: SFT on Gemma3-12B/27B, 2: smaller models and 3: other peft approaches). We will also revise the main text paper, including some of the experiments above in the main figures, taking into account the clarity comments raised by reviewer H2hK, and clarifying our discussion section.
>
> We thank all the reviewers again for their constructive feedback, which helped us strengthen our work. We will follow up with more individual responses.
>
> Edit: fixed figure numbers following upload of revised manuscript

---

### Author Response · Authors · 2025-12-03
**Final comment for AC**

Dear AC,

Thank you for your service in these difficult circumstances. As per the guidelines, here is a brief summary of our paper and of the discussion with reviewers.

Our core contributions are:
- A comparison of ICL and SFT, viewed as two learning algorithms, as they learn classification tasks with increasing numbers of examples (shots). This controlled, head-to-head comparison in medium-sized language models reveals quantitative differences in inductive biases between the two learning algorithms.
- We study the change of internal model representations across layers and learning trajectories. We find that SFT consistently collapses representation along task labels, whereas ICL preserves richer representations, mirroring the different inductive biases of the two learning algorithms. This finding is consistent across models, and may explain recent reports of ICL’s superior generalization compared to SFT.

The reviewers were overall positive of our work, praising its presentation -“an excellent job of clearly presenting its results” (Reviewer b4Qy), “well-situated within the broader literature” (Reviewer b4Qy). Our comparison of ICL and SFT was deemed a “useful and interesting framing” (Reviewer sBMX), addressing “a broad, pressing question in the field” (Reviewer fv6R). Our 2D linear task enabled a “matched‑trajectory setup [that] isolates algorithmic differences between ICL and SFT more cleanly than typical open‑domain benchmarks” (Reviewer sBMX), while the RSA analysis “add[s] a strong layer of evidence beyond simple task performance metrics” (Reviewer b4Qy), in a “compelling” and “cool” way (Reviewer sBMX). The breadth of the models tested was also appreciated (Reviewer H2hK & fv6R).

Reviewers made numerous suggestions to strengthen the contribution, which triggered many additional experiments on our side. These new experiments are in the appendix (Supp. Figs 10-20), please see the common comment below for a description. We believe to have addressed all the points raised in the reviews, and the only reviewer who was still hesitant about acceptance raised their score following our rebuttal, while the others did not have the opportunity to interact further with us.

Sincerely,
The authors

---

### Meta-Review · Area_Chair_YxGW · 2025-12-23

**Summary:**

**Paper Summary:**
This paper empirically compares the learning dynamics of in-context learning (ICL) and supervised fine-tuning (SFT) in mid-sized language models on 2D numerical and semantic classification tasks. Although the two approaches achieve similar performance, the study shows that they exhibit distinct inductive biases: ICL tends to preserve pretraining priors, whereas SFT can be brittle. An analysis of intermediate representations further reveals a feature collapse phenomenon in SFT. These findings are consistently observed across different foundation models and tasks.

**Concerns:**
Reviewers raised concerns about the limited experimental scope, novelty relative to prior work, the need for additional experiments on smaller models, insufficient clarification of experimental settings and claims, and the lack of evaluations on more tasks and foundation models.

**Recommendation:**
The authors have adequately addressed the reviewers’ concerns. The paper is clearly presented and provides deeper insight into both the differences and similarities between ICL and SFT. Therefore, the AC recommends **acceptance**.

**Reviewer Concerns:**

Reviewers raised concerns about the limited experimental scope, novelty relative to prior work, the need for additional experiments on smaller models, insufficient clarification of experimental settings and claims, and the lack of evaluations on more tasks and foundation models. The authors have adequately addressed the reviewers’ concerns.

**Reviewer Scores:**

The reviewers’ scores changed after the rebuttal, given the discussion context, from 6, 6, 6, 4 $\rightarrow$ 6, 6, 6, 6.

---

### Decision · Program_Chairs · 2026-01-26

Accept (Poster)